# A dual role for SAMHD1 in regulating HBV cccDNA and RT-dependent particle genesis

Peter AC Wing[1], Tamara Davenne[2], Jochen Wettengel[3], Alvina G Lai[1], Xiaodong Zhuang[1], Anindita Chakraborty[3], Valentina D'Arienzo[1], Catharina Kramer[1], Chunkyu Ko[3], James M Harris[1], Sabrina Schreiner[3,4], Martin Higgs[5], Stephanie Roessler[6], Joanna L Parish[5], Ulrike Protzer[3,4], Peter Balfe[5], Jan Rehwinkel[2], Jane A McKeating[1]

Chronic hepatitis B is one of the world's unconquered diseases with more than 240 million infected subjects at risk of developing liver disease and hepatocellular carcinoma. Hepatitis B virus reverse transcribes pre-genomic RNA to relaxed circular DNA (rcDNA) that comprises the infectious particle. To establish infection of a naïve target cell, the newly imported rcDNA is repaired by host enzymes to generate covalently closed circular DNA (cccDNA), which forms the transcriptional template for viral replication. SAMHD1 is a component of the innate immune system that regulates deoxyribonucleoside triphosphate levels required for host and viral DNA synthesis. Here, we show a positive role for SAMHD1 in regulating cccDNA formation, where KO of SAMHD1 significantly reduces cccDNA levels that was reversed by expressing wild-type but not a mutated SAMHD1 lacking the nuclear localization signal. The limited pool of cccDNA in infected *Samhd1* KO cells is transcriptionally active, and we observed a 10-fold increase in newly synthesized rcDNA-containing particles, demonstrating a dual role for SAMHD1 to both facilitate cccDNA genesis and to restrict reverse transcriptase-dependent particle genesis.

## Introduction

Chronic hepatitis B is one of the world's most economically important diseases, with 2 billion people exposed to the virus at some stage of their lives. Hepatitis B virus (HBV) replicates in the liver, and chronic infection can result in progressive liver disease, cirrhosis, and hepatocellular carcinoma. HBV is the third leading cause of cancer-related deaths, with an estimated mortality of 695,000 deaths per year (Ringelhan et al, 2017). HBV is the prototypic member of the hepadnaviruses, a family of small enveloped hepatotropic viruses with a partial double-stranded relaxed circular DNA (rcDNA) genome. Following infection, the rcDNA is imported to the nucleus and converted to covalently closed circular DNA (cccDNA) that serves as the transcriptional template for viral RNAs. The rcDNA represents the mature form of the viral genome that is packaged into nucleocapsids that are enveloped and released as newly formed infectious virions or redirected toward the nucleus to replenish and maintain the pool of episomal cccDNA. This amplification pathway, together with the long half-life of cccDNA contributes to viral persistence (Urban et al, 2010; Ko et al, 2018). HBV does not require integration into the host genome for replication; however, integrated viral DNA fragments are commonly found in chronic hepatitis B and may contribute to carcinogenesis (Tu & Urban, 2018).

The mechanisms underlying HBV rcDNA repair and early steps in cccDNA formation are not well defined (Schreiner & Nassal, 2017) and several members of the host DNA repair pathway are reported to play a role. Tyrosyl-DNA phosphodiesterase 2 (TDP-2) cleaves the topoisomerase-like linkage between the polymerase and rcDNA (Koniger et al, 2014; Cui et al, 2015); flap endonuclease (FEN1) excises the overlapping regions in rcDNA (Kitamura et al, 2018) together with the polymerases κ and λ (Qi et al, 2016) and ligases LIG1 and LIG3 (Long et al, 2017) that repair and ligate the incomplete rcDNA regions, respectively. HBV cccDNA copy number in the chronically infected liver, in vitro culture systems, and infected chimeric liver mice is low (Werle-Lapostolle et al, 2004; Volz et al, 2013; Nassal, 2015) and not affected by the currently used nucleoside and nucleotide analogue therapies that only suppress HBV replication. Hence, a greater understanding of the host pathways regulating HBV cccDNA formation will aid the development of curative treatments that will eliminate or permanently silence this episomal DNA reservoir.

Sterile alpha motif and histidine–aspartic acid domain containing protein 1 (SAMHD1) is a deoxyribonucleoside triphosphate (dNTP) triphosphohydrolase (Goldstone et al, 2011; Powell et al,

[1]Nuffield Department of Medicine, University of Oxford, Oxford, UK   [2]Medical Research Council Human Immunology Unit, Medical Research Council Weatherall Institute of Molecular Medicine, Radcliffe Department of Medicine, University of Oxford, Oxford, UK   [3]Institute of Virology, Technische Universität München/Helmholtz Zentrum München, Munich, Germany   [4]German Center for Infection Research (DZIF), Munich Partner Site, Munich, Germany   [5]Institutes of Cancer and Genomic Sciences and Immunity and Immunotherapy, College of Medical and Dental Sciences, University of Birmingham, UK   [6]Institute of Pathology, University Hospital Heidelberg, Heidelberg, Germany

Correspondence: jane.mckeating@ndm.ox.ac.uk

2011) that restricts HIV-1 infection of myeloid cells and CD4[+] T cells by depleting dNTPs required for reverse transcription (Hrecka et al, 2011; Laguette et al, 2011; Baldauf et al, 2012; Lahouassa et al, 2012). HBV replication is dependent on reverse transcription during a late step in its life cycle where encapsidated pre-genomic RNA (pgRNA) is converted to rcDNA by the viral encoded polymerase (Urban et al, 2010). Sommer et al reported a restrictive role for SAMHD1 in HBV reverse transcription where siRNA knockdown (KD) induced a modest twofold increase in secreted HBV particles (Sommer et al, 2016). Viruses generally evolve to evade or counteract host restriction factors, for example, lentiviral accessory proteins Vpx and Vpr target SAMHD1 for proteasomal degradation via the E3 cellular ubiquitin ligase complex (Laguette et al, 2011; Lim et al, 2012). HBV infection does not degrade SAMHD1 (Sommer et al, 2016), suggesting additional roles for SAMHD1 in the viral life cycle.

Mutations in SAMHD1 are implicated in cancer development (Mauney & Hollis, 2018) and in a severe congenital inflammatory disease known as Aicardi–Goutières syndrome that is characterized by an overproduction of type I IFNs (Crow & Manel, 2015). Recent reports highlight a role for SAMHD1 to regulate DNA repair independent of its dNTPase activity (Goncalves et al, 2012; Tungler et al, 2013; Seamon et al, 2015; Seamon et al, 2016; Daddacha et al, 2017). SAMHD1 can interact with the Mre11 exonuclease and stimulate its activity (Coquel, Neumayer et al, 2018a), suggesting a scaffolding role for SAMHD1 in coordinating the activity of the DNA replisome (Pasero & Vindigni, 2017). In SAMHD1-depleted cells, single-stranded DNA fragments are released from stalled replication forks and accumulate in the cytosol, where they activate the cyclic GMP-AMP synthase (cGAS)–stimulation of interferon genes (STING) pathway to induce expression of pro-inflammatory type I IFNs (Coquel, Silva et al, 2018b).

Here, we show two distinct roles for SAMHD1 in the HBV life cycle. First, SAMHD1 is required during the early steps of cccDNA formation. Second, using *Samhd1* KO hepatoma cells, we observed a significant 1–2 log increase in newly synthesized and secreted HBV DNA particles. Our studies highlight a dual role for SAMHD1, both as a host-dependency factor and as a restriction factor, depending on the stage in the HBV life cycle.

# Results

## SAMHD1 positively regulates HBV cccDNA levels

HBV preferentially infects nondividing hepatocytes in the liver and in vitro culture systems use DMSO to both arrest cells and induce the expression of liver-specific differentiation factors required for viral replication. We used HepG2-NTCP (K7) cells as they support efficient HBV replication (Ko et al, 2018) and express basal levels of SAMHD1 and its phosphorylated form (P-SAMHD1) (Fig 1A). We confirm that SAMHD1 is induced by IFNs, consistent with previous reports (Berger et al, 2011) (Fig 1A). Moreover, screening a panel of human hepatocyte-derived cells showed that all lines expressed basal levels of SAMHD1 in the absence of IFN treatment (Fig S1). Thus, HepG2-NTCP cells recapitulate the IFN-dependent regulation of SAMHD1 reported in the literature.

The Vpx proteins of the SIV$_{SMM}$/HIV-2 lentiviruses bind SAMHD1 and recruit the E3 ubiquitin ligase that targets SAMHD1 for proteasomal degradation (Laguette et al, 2012; Lim et al, 2012). To exploit this, we used lentivirus-like particles (LVPs) containing Vpx

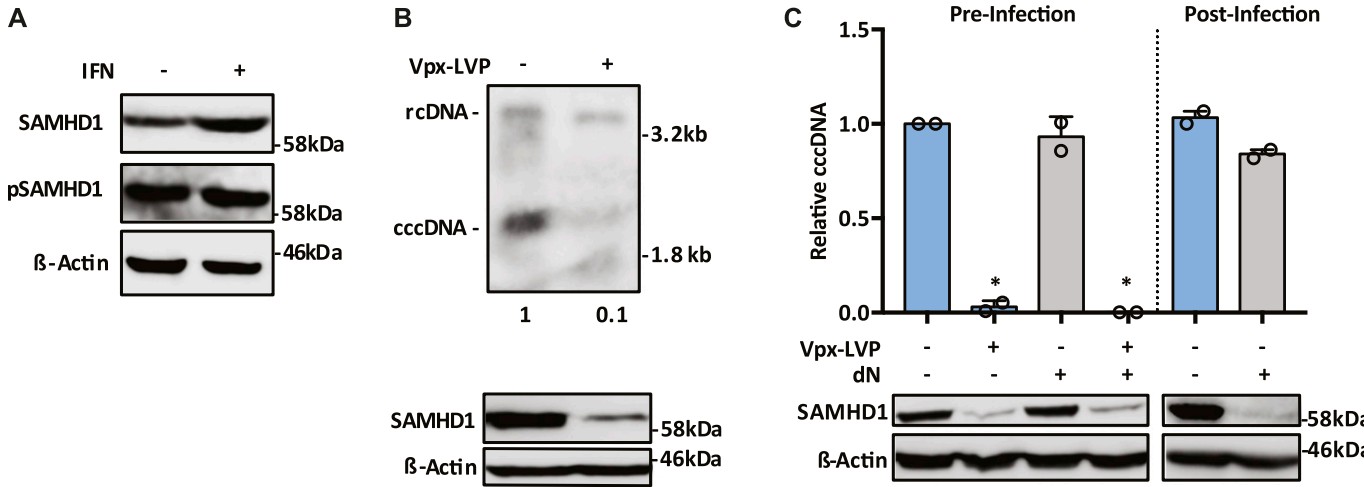

**Figure 1.  Vpx-targeted degradation of SAMHD1 in HepG2-NTCP reduces HBV cccDNA levels.**
**(A)** HepG2-NTCP cells were arrested with 2% DMSO 3 d before treating with IFNa (1,000 IU/ml) for 24 h. Levels of total and phosphorylated SAMHD1 were assessed by Western blot. **(B)** HepG2-NTCP cells were transduced with Vpx-LVPs and infected with HBV at an MOI of 200 for 6 d. DNA extracted after HIRT lysis was analyzed by Southern blotting using an HBV-specific probe. Size markers of a known length were run as a control. Numbers below the blot indicate densitometric values for cccDNA ± Vpx-LVP transduction. Vpx knockdown of SAMHD1 was confirmed by Western blotting. **(C)** HepG2-NTCP cells were transduced with Vpx-LVPs or supplemented with 0.5 mM dN 24 h before infection with HBV at an MOI of 200 genome equivalents (GEs). cccDNA levels were quantified at 3 d postinfection by qPCR. In addition, HepG2-NTCP cells were infected as before with Vpx-LVPs delivered 3 d postinfection. The cells were cultured for a further 3 d and harvested for cccDNA quantification by qPCR. Data represent two independent experiments, each comprising three replicates per condition and are presented as mean ± SEM. Statistical analysis was performed using a Mann–Whitney *U* test (*$P ≤ 0.05$).
Source data are available for this figure.

to degrade SAMHD1 at defined times to investigate its role in HBV replication. Vpx reduced SAMHD1 expression in HepG2-NTCP, and infecting these cells with HBV resulted in significantly lower levels of cccDNA as measured by Southern blotting and PCR (Fig 1B and C), suggesting that SAMHD1 may play a pro-viral role in the establishment of the HBV transcription template. Importantly, providing exogenous pyrimidine and purine deoxynucleosides (dNs) to HepG2-NTCP cells had no effect on cccDNA levels, suggesting that HBV cccDNA formation is not limited by dNTP levels (Fig 1C). To establish when SAMHD1 regulates HBV cccDNA levels, we infected cells with HBV and 3 d later, transduced with LVP-Vpx and measured cccDNA levels after a further 3 d (i.e., 6 d post HBV infection). Although the KD of SAMHD1 was efficient in the HBV-infected cells, we observed a negligible effect on cccDNA levels (Fig 1C), suggesting

that SAMHD1 is required during the early stages of cccDNA formation.

To extend these observations, we used CRISPR-Cas9 to KO SAMHD1 in HepG2 NTCP cells. Western blotting (Fig 2A) and genome sequencing identified two KO clones U8 and U12. In line with previous reports, we demonstrate increased dNTP levels in both KO clones compared with Wt cells (Fig 2B), with a significant increase in dGTP compared with other dNTPs (Franzolin et al, 2013). Moreover, to confirm that *Samhd1* KO cells were deficient in regulating DNA resection during double-stranded break (DSB) repair (Daddacha et al, 2017), we treated Wt and KO (clone 8) HepG2 NTCP cells with the topoisomerase inhibitor camptothecin (CPT) to induce DSBs and assessed resection levels. These analyses showed a significant reduction in the level of phosphorylated RPA32 foci, a marker of resected DNA after DSB repair, in KO

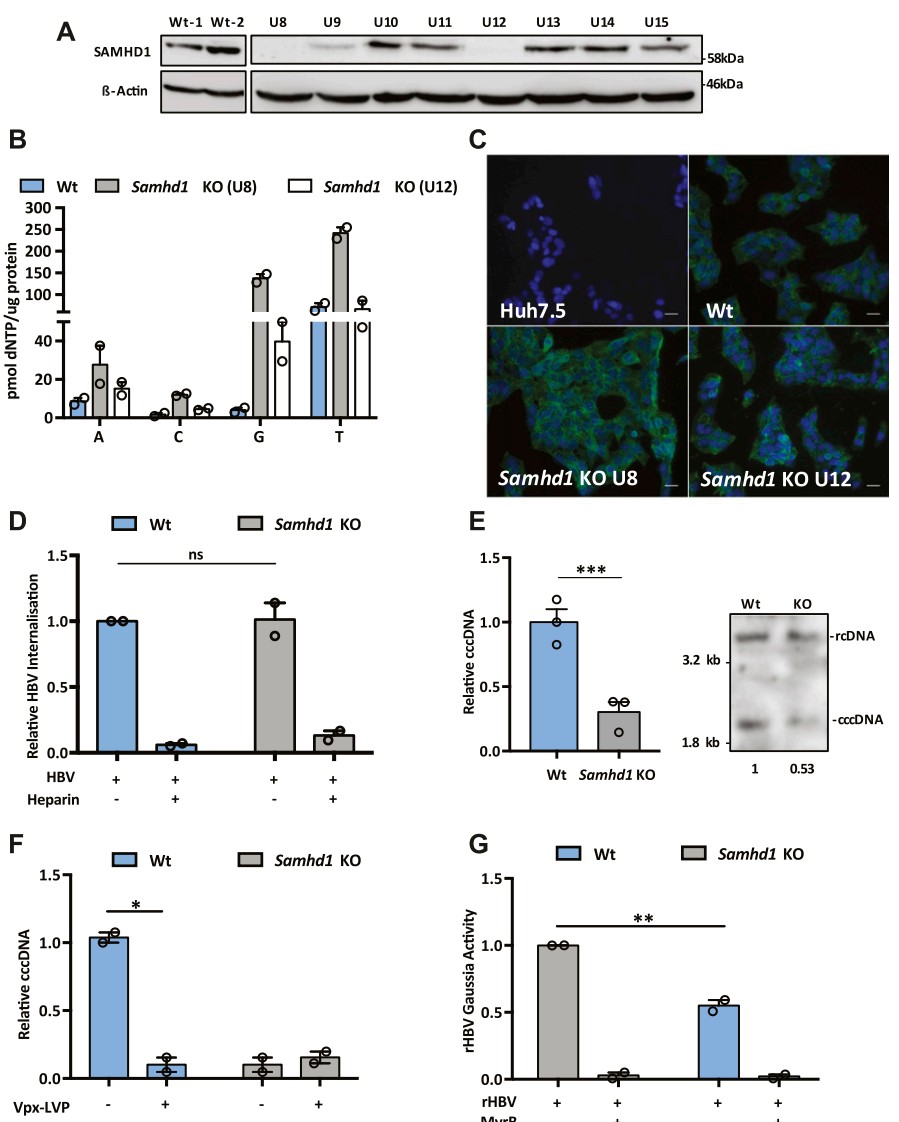

**Figure 2. *Samhd1* KO in HepG2-NTCP cells reduces HBV cccDNA.**
**(A)** HepG2-NTCP cells were transfected with guide RNAs for CRISPR-Cas9 KO of *Samhd1* and single colonies obtained by limiting dilution. The colonies were amplified and screened for SAMHD1 expression by Western blot. **(B)** Extracts were prepared from parental and *Samhd1* KO clones U8 and U12 and dNTP levels measured. Data are shown relative to total cellular protein and represent two independent experiments, each comprising three replicates per condition and presented as mean ± SEM. **(C)** NTCP expression in *Samhd1* KO HepG2-NTCP clones U8 and U12 was tested using Myrcludex B (MyrB) tagged with an Alexa 488 fluorophore. Huh-7.5 cells (NTCP negative) were used as a control. Images were taken on a 20× objective; scale bar depicts 20 μm. **(D)** HBV uptake in Wt and *Samhd1* KO U8. HepG2-NTCP Wt and *Samhd1* KO U8 cells were infected with HBV for 6 h using a synchronized infection protocol (see the Materials and Methods section) and total HBV DNA levels quantified by qPCR. Data are shown relative to Wt cells. As a control to block HBV entry, the cells were treated with heparin (50 IU/ml) for 1 h before infection. Data represent two independent experiments, each comprising three replicates per condition and are presented as mean ± SEM. Statistical analysis was performed using a Mann–Whitney *U* test. **(E)** HepG2-NTCP Wt and *Samhd1* KO U8 cells were infected with HBV at an MOI of 200 for 6 h and cccDNA levels determined at 3 d postinfection by qPCR. Data are shown relative to Wt cells and represent three independent experiments, each comprising three replicates per condition and are presented as mean ± SEM. Statistical analysis was performed using a Mann–Whitney *U* test (***P ≤ 0.001). DNA extracted from infected Wt and SAMHD1 KO cells was analyzed by Southern blotting using an HBV-specific probe. Size markers of a known length were run as a control. Numbers below the blot indicate densitometric values for cccDNA in Wt and KO cells. **(F)** Parental and *Samhd1* KO U8 cells were transduced with Vpx-LVP 24 h before HBV infection. The cells were infected with HBV at an MOI of 200 for 3 d after which cccDNA was quantified by qPCR. Data represent two independent experiments, each comprising three replicates per condition and presented as mean ± SEM. Statistical analysis was performed using a Mann–Whitney *U* test (*P ≤ 0.05). **(G)** Infection of Wt and *Samhd1* KO U8 cells with a single-cycle HBV-Gaussia

reporter virus (HBV-Gluc). The cells were infected with HBV-Gluc at an MOI of 200 and luciferase activity measured 24 h postinfection. The cells treated with the entry inhibitor MyrB (1 μM) acted as a control for background luciferase signal. Data represent two independent experiments, each comprising three replicates per condition and are presented as mean ± SEM. Statistical analysis was performed using a Mann–Whitney *U* test (**P ≤ 0.01).
Source data are available for this figure.

cells compared with Wt (Fig S2). Because Sodium Taurocholate Cotransporting Polypeptide (NTCP) is essential for HBV entry, we confirmed that both *Samhd1* KO clones express the viral receptor using a fluorescent-labelled HBV Pre-S1 peptide mimetic that binds NTCP (Myrcludex B) (Fig 2C). Thus, our KO cells recapitulate the reported phenotype of cells depleted of SAMHD1 and provide an ideal model for studying its role in the HBV life cycle.

We used a recently developed synchronized infection assay that measures HBV cellular uptake (Ko et al, 2018) to determine the effect of SAMHD1 on viral entry. We noted comparable levels of virus internalization into Wt and *Samhd1* KO cells during a short-term 6-h infection assay (Fig 2D). As a positive control, we demonstrated that heparin blocked the viral uptake as previously reported (Ko et al, 2018; Schulze et al, 2007). Importantly, we noted a significant reduction in cccDNA levels in the *Samhd1* KO cells compared to Wt at 3 d postinfection (Figs 2E and S3), consistent with our earlier data using Vpx-mediated degradation of SAMHD1. To exclude the possibility that Vpx may have additional effects on host cellular pathways, we transduced *Samhd1* KO cells with Vpx-LVPs and reassuringly this showed no additional effect on cccDNA levels (Fig 2F). In an effort to study the role of SAMHD1 in cccDNA formation at earlier time points following infection, we used a recombinant HBV reporter virus encoding Gaussia luciferase (rHBV-Gaussia) where luminescence is dependent on cccDNA formation (Untergasser & Protzer, 2004). We observed a significant reduction in Gaussia activity in the *Samhd1* KO cells at 24 h postinfection (Fig 2G), confirming our earlier observations that SAMHD1 positively regulates early steps in cccDNA formation and/or maintenance.

SAMHD1 is a nuclear protein that preferentially uses karyopherin $\alpha$2 and a classical N-terminal nuclear localization signal (NLS) to traffic to the nucleus (Schaller, Pollpeter et al, 2014a). To investigate the role of SAMHD1 nuclear localization in HBV cccDNA genesis, we transduced *Samhd1* KO cells with lentiviruses expressing HA-tagged Wt or K11A SAMHD1 mutated in the NLS (Schaller et al, 2014a). We confirmed cytoplasmic localization of SAMHD1$_{K11A}$ (Fig 3A) and a significant reduction in cellular dGTP and dATP levels, confirming catalytic activity of the NLS mutant (Fig 3B). Importantly, Wt SAMHD1 restored the cccDNA copy number to comparable levels seen in Wt HepG2-NTCP cells; however, SAMHD1$_{K11A}$ had a minimal effect on cccDNA levels (Fig 3C), demonstrating a role for nuclear SAMHD1 in the early steps of HBV cccDNA formation.

SAMHD1 can promote the degradation of nascent DNA at stalled replication forks, resulting in the accumulation of single-stranded DNA fragments that activate the cGAS–STING pathway and induce IFNs (Coquel et al, 2018b). APOBEC3 (A3) cytidine deaminases are a family of IFN stimulated enzymes that can degrade cccDNA and inhibit HBV replication (Lucifora et al, 2014). We investigated whether HBV infection of Wt or *Samhd1* KO cells induced the expression of APOBECs along with MxA and ISG20 following a 6-h synchronized infection. Interestingly, we observed a significant increase in MxA, ISG20, and APOBEC mRNA levels in HBV-infected KO cells compared with Wt (Fig 4) that was dependent on HBV internalization, suggesting that SAMHD1 acts to suppress innate immune activation.

### SAMHD1 restricts de novo HBV particle genesis

HBV replication is defined by the size and transcriptional output of the cellular cccDNA pool, and we were interested to know whether

SAMHD1 could regulate cccDNA transcriptional activity. PCR quantification of pgRNA levels at 6 d postinfection of Wt, and *Samhd1* KO cells showed comparable levels of viral transcripts relative to cccDNA levels in both cell types (Fig S4), suggesting a minimal role for SAMHD1 in modulating cccDNA epigenome and associated transcriptional activity.

SAMHD1 was previously reported to limit the de novo genesis of encapsidated rcDNA particles, where siRNA silencing of SAMHD1 induced a modest twofold increase in the secretion of HBV DNA from the stable HBV producer line HepG2.2.15 (Chen et al, 2014; Sommer et al, 2016). We show that Vpx-mediated KD of SAMHD1 in HepG2.2.15 resulted in a 20-fold increase in extracellular HBV DNA levels when the cells were pretreated with the nucleoside analog entecavir (ETV) (Fig 5A). In contrast, we observed a modest non-significant effect of SAMHD1 KD in HepG2.2.15 cells without prior ETV treatment, most likely reflecting the accumulation and long half-life of intracellular encapsidated rcDNA genomes in this producer cell line (Fig 5A). We confirmed that treating HepG2.2.15 cells with ETV reduced extracellular HBV DNA levels and removing the drug by extensive washing reversed this inhibition and allowed one to observe reverse transcriptase (RT) dependent particle genesis (Fig S5). Applying this ETV pretreatment and "wash-out" protocol to an independent HBV producer line HepAD38 (Ladner et al, 1997) showed that Vpx-mediated KD of SAMHD1 induced a 10-fold increase in HBV DNA levels (Fig S6). We sought to validate these results in de novo–infected HepG2-NTCP cells and found that Vpx-mediated degradation of SAMHD1 significantly increased the levels of extracellular HBV DNA above the ETV control (Fig 5B). Adenoviral vectors encoding HBV genomes (Ad-HBV) have been used extensively to deliver the HBV transcriptional template to cells and support robust particle assembly (Huang et al, 2012), allowing us to focus on the late steps of the viral life cycle. Ad-HBV-GFP–transduced HepG2-NTCP cells secrete HBV DNA in an RT-dependent manner (Fig S7), and Vpx-KD of SAMHD1 resulted in the secretion of 11-fold more particles (Fig 5C). Furthermore, repeating this experiment in our *Samhd1* KO cells showed a robust phenotype with a 100-fold increase in secreted HBV particles enabling us to quantify particle production over time (Fig 5D). In line with reports that SAMHD1 restricts HIV replication, we confirm a significant increase in HIV RT-dependent replication in our *Samhd1* KO hepatoma cells (Fig S8). In summary, we show that SAMHD1 restricts the genesis of HBV DNA-containing particles.

## Discussion

Our study highlights a dual role for SAMHD1 in the HBV life cycle, a positive role in regulating cccDNA levels and a restriction of polymerase-mediated reverse transcription of pgRNA to rcDNA. Since our experiments focus on the first 3 d of infection and cccDNA is long-lived with an estimated half-life of 40 d in vitro (Ko et al, 2018), we are unable to assess the role of SAMHD1 in cccDNA maintenance. To the best of our knowledge, this is the first observation describing a pro-viral role for SAMHD1 and is consistent with reports on the role of SAMHD1 in DNA repair (Coquel et al, 2018a). SAMHD1 phosphorylation, mediated by cyclin-dependent

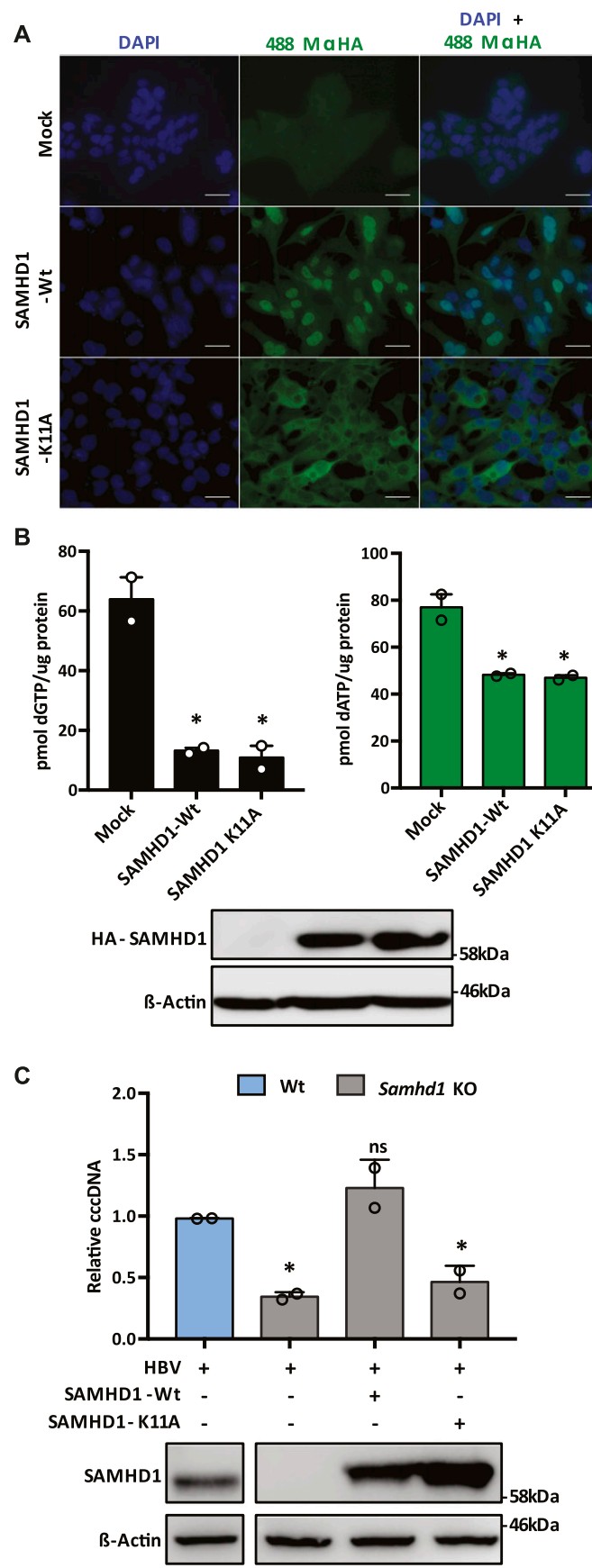

**Figure 3. Nuclear localization of SAMHD1 is required for HBV cccDNA genesis.**
**(A)** Immunofluorescent images of *Samhd1* KO clone U8 cells transduced with lentivirus expressing HA-tagged SAMHD1 Wt or K11A mutant . Images were taken using an X63 objective; scale bars indicate 50 $\mu$m. **(B)** HepG2-NTCP *Samhd1* KO U8 cells were transduced with lentivirus expressing HA-tagged Wt or SAMHD1$_{K11A}$ for 24 h. Expression was confirmed by Western blotting and dNTP levels quantified. dNTP levels were normalized to total cellular protein. Data from two independent experiments are shown as mean ± SEM. Statistical analysis was performed using a Mann–Whitney $U$ test (*$P \leq 0.05$). **(C)** HepG2-NTCP *Samhd1* KO U8 cells were transduced with lentivirus expressing Wt or SAMHD1$_{K11A}$ and 24 h later infected with HBV at an MOI of 200. cccDNA was quantified by qPCR at 3 d postinfection and SAMHD1 expression confirmed by Western blotting. Data represent two independent experiments, each comprising three replicates per condition and are presented as mean ± SEM. Statistical analysis was performed using a Mann–Whitney $U$ test (*$P \leq 0.05$).
Source data are available for this figure.

kinases CDK1 and CDK2 (St Gelais, 2014), is required for the resection of nascent DNA strands at stalled replication forks (Daddacha et al, 2017) and may act as a switch between the nuclease and dNTPase functions that regulate HBV cccDNA genesis and de novo particle genesis, respectively (Fig 6).

Upon infection, the HBV capsid is transported to the nucleus where the rcDNA is released and repaired to generate cccDNA. HBV rcDNA has both strand breaks and single-stranded regions that require trimming, elongation, and ligation, and several members of the DNA damage response pathway have been reported to mediate these processes (Schreiner & Nassal, 2017). The identification of alternative enzymes for many of the functions required for cccDNA genesis suggests functional redundancy within the DNA repair system (reviewed in Mayanagi et al (2018)). SAMHD1 can bind ssDNA and has been reported to act as a scaffolding protein to promote both homologous recombination and DNA resection (Goncalves et al, 2012; Beloglazova et al, 2013; Tungler et al, 2013; Seamon et al, 2015, 2016; Daddacha et al, 2017), functions that may account for its stimulatory role in the rcDNA repair process. At the present time, our experiments cannot discriminate whether SAMHD1 regulates the half-life of incoming rcDNA particles or specifically binds HBV DNA template. Our observation that mutation of the NLS of SAMHD1 did not restore cccDNA levels in HBV-infected *Samhd1* KO cells, while modulating cellular dNTP levels, demonstrate a requirement for nuclear SAMHD1 that is independent of its dNTPase activity. This observation is consistent with reports that SAMHD1-depleted cells show impaired DNA replication fork progression (Franzolin et al, 2013; Rentoft et al, 2016).

The innate immune system is the first line of host defense against a broad range of pathogens, including viruses and cellular stress by sensing damage-associated molecular patterns, including cytosolic DNA. This DNA is recognized by a variety of nucleic acid sensors that converge on STING to transactivate innate immune response genes (Dhanwani et al, 2018). A key player in this process is cGAS that is triggered by cytosolic DNA accumulating after ionizing radiation (Hartlova et al, 2015) or by exposure to agents targeting DNA replication forks (Yang et al, 2007; Lam et al, 2014; Shen et al, 2015). Importantly, we show that HBV infection of *Samhd1* KO cells induces several genes of the innate immune system (MxA, ISG20, and APOBECs), and it is noteworthy that APOBEC3A and 3B have been reported to degrade cccDNA levels in HBV-infected hepatocytes (Lucifora et al, 2014). The negligible induction of any

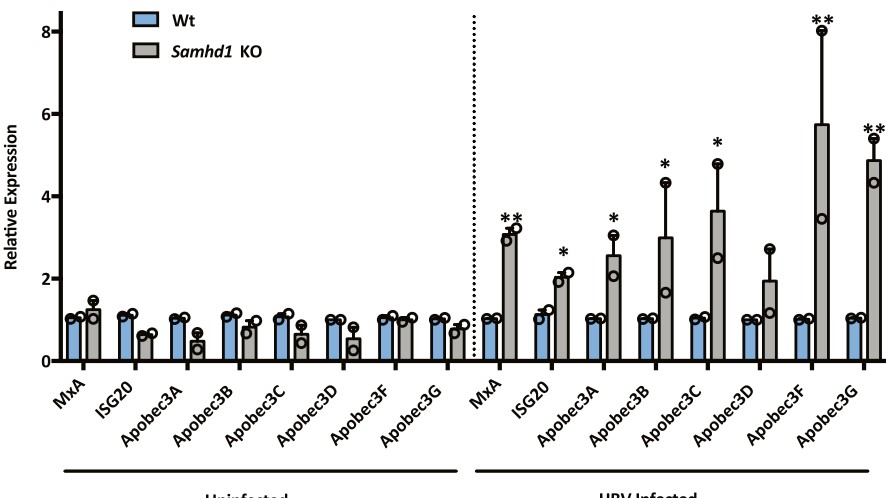

**Figure 4.  HBV activation of innate immune responses in *Samhd1* KO cells.**
*MxA, ISG20, Apobec 3A, 3B, 3C, 3D, 3F,* and *3G* RNA levels were measured by RT-qPCR in Wt and *Samhd1* KO U8 cells that were infected with HBV for 6 h using a synchronized infection protocol (Fig 2D). As a control, transcript levels were measured in heparin-treated cells that blocked HBV internalization. Data represent two independent experiments, each comprising three replicates per condition and are presented as mean ± SEM. Statistical analysis was performed using a Mann–Whitney *U* test (*$P$ ≤ 0.05, **$P$ ≤ 0.01).

ISGs in HBV-infected Wt HepG2-NTCP cells is consistent with earlier reports describing HBV as a "stealth" virus that does not induce IFN responses in chimpanzees (Wieland et al, 2004), acutely infected patients (Dunn et al, 2009; Stacey et al, 2009), or in vitro infection models (Cheng et al, 2017; Niu et al, 2017; Mutz et al, 2018). These results highlight a potential new role for SAMHD1 in regulating host innate recognition of HBV genomes that may restrict the cccDNA reservoir in vivo. The interplay between DNA damage response factors, including SAMHD1, and innate immune sensors is not well defined and further studies to define pathways regulating the

stability of incoming HBV rcDNA particles and trafficking to the nucleus are required to understand the complex mechanisms underlying cccDNA genesis and maintenance.

Using a variety of HBV model systems, we demonstrate that Vpx-targeted degradation or genetic KO of SAMHD1 results in a significant 10-fold to 100-fold increase in secreted HBV DNA. This contrasts with the modest twofold increase in secreted encapsidated viral DNA previously reported (Chen et al, 2014; Jeong et al, 2016; Sommer et al, 2016). These differences may reflect the use of siRNA that only partially suppressed *Samhd1 m*RNA levels or the

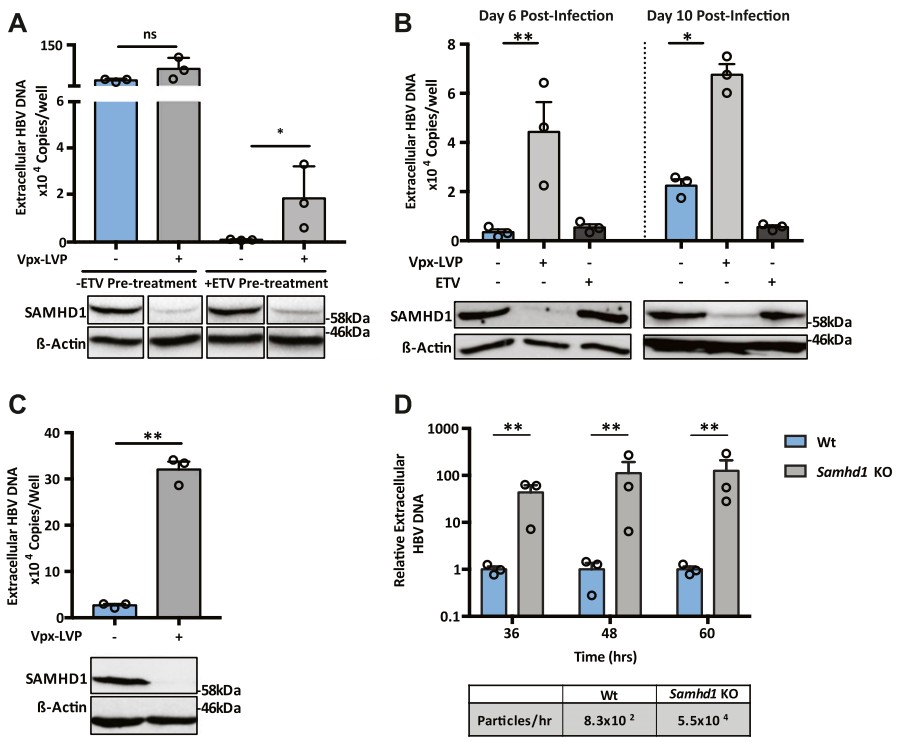

**Figure 5.  SAMHD1 restricts HBV particle genesis.**
**(A)** HepG2.2.15 cells were pretreated with ETV (1 μM) or vehicle control for 72 h, drug was removed by washing, and cells transduced with Vpx-LVPs. The cells were cultured for 72 h and extracellular HBV DNA quantified by qPCR. Data represent three independent experiments that comprised three technical replicates and are presented as mean ± SEM. SAMHD1 expression is confirmed by Western blotting. **(B)** HepG2-NTCP cells were transduced with Vpx-LVPs 24 h before infecting with HBV at an MOI of 200. Secreted HBV DNA was measured at 6 and 10 d postinfection. As before, to control for nascent particle production, the cells were treated with 1 μM of entecavir (ETV). Data represent three independent experiments that comprised three technical replicates and are presented as mean ± SEM. Statistical analysis was performed using a Mann–Whitney *U* test (*$P$ value ≤ 0.05, **$P$ value ≤ 0.01). **(C)** HepG2-NTCP cells were transduced with Vpx-LVP and 24 h later, infected with Ad-HBV-GFP at an MOI of 20. After 3 d, secreted HBV DNA was measured and SAMHD1 expression assessed by Western blot. Transduced cells were treated with 1 μM of ETV as a control for nascent particle production (data not shown). Data represent three independent experiments and are presented as mean ± SEM. Statistical analysis was performed using a Mann–Whitney *U* test (**$P$ value ≤ 0.01). **(D)** Wt and *Samhd1* KO cells were transduced with Ad-HBV-GFP at an MOI of 20 and extracellular HBV DNA was measured at indicated time points. Particle production per hour was estimated and values are shown below the graph. Data represent three independent experiments that

comprised three technical replicates and are presented as mean ± SEM relative to the Wt control. Statistical analysis was performed using a Mann–Whitney *U* test (**$P$ value ≤ 0.01). Source data are available for this figure.

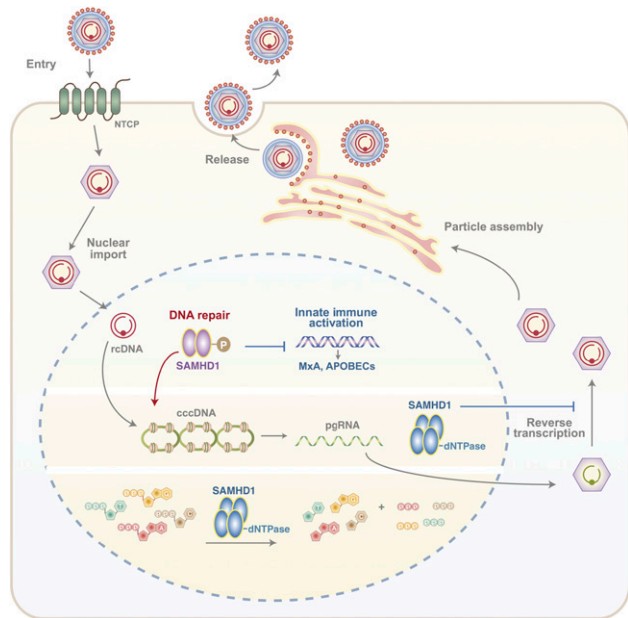

**Figure 6. Proposed role of SAMHD1 in HBV life cycle.**
Phosphorylated SAMHD1 plays a key role in HBV rcDNA to cccDNA conversion in the nucleus of an infected cell as well as regulating innate immune activation genes such as MxA and APOBECS. In addition, SAMHD1 dNTPase restricts the ability of viral polymerase to reverse transcribe pgRNA-rcDNA, required for the genesis of new infectious particles.

overexpression of SAMHD1 that may perturb the balance of cellular dNTPs and compromise data interpretation. We noted that pre-treating HBV producer cells with ETV to reduce intracellular levels of encapsidated rcDNA before targeting SAMHD1 was required to observe a significant increase in secreted HBV DNA and so this may also contribute to the modest phenotype reported in earlier studies. Our observation that SAMHD1 regulates the production of infectious HBV particles is of interest, given recent reports that de novo particle transmission events may be required to maintain the cccDNA pool (Goyal et al, 2017; Allweiss et al, 2018; Giersch et al, 2019).

Viruses evolve sophisticated means to evade or directly counteract restriction factors. Our infection experiments have not shown any perturbation in SAMHD1 expression levels, consistent with a previous report (Sommer et al, 2016), which is unsurprising, given the key role of SAMHD1 in cccDNA formation. Of note, HBV infection has been reported to induce ribonucleotide reductase subunit 2 (RNR-R2) expression via an HBx-p53–dependent pathway and thereby increase cellular dNTP levels (Cohen et al, 2010; Ricardo-Lax et al, 2015). We confirmed that HBV-induced RNR-R2 expression led to an increase in cellular dNTPs (Fig S9); however, this was not sufficient to overcome SAMHD1 restriction of new particle genesis. Because SAMHD1 activity is regulated via cell cycle–dependent kinases, HBV infection may alter cell cycle and thereby modulate SAMHD1 restriction. A number of viruses, including DNA viruses, RNA viruses, and retroviruses, induce cell cycle arrest between the G2 and M phases, allowing viral genomes to replicate before the cell enters mitosis. Human papillomavirus induces G2/M phase arrest via E4 maintenance of Cdk1 phosphorylation (Knight et al, 2006). Hepatitis C virus induces G2/M arrest of hepatocytes via antioxidative stress and TGF signaling (Walters et al, 2009; Kannan et al,

2011). Xia et al recently reported that HBV infection arrests hepatocytes in G2/M cycle (Xia et al, 2018); however, this contrasts with previous reports showing that HBV replication is independent of the cell cycle in a transgenic mouse model (Guidotti et al, 1997). HBx protein has been implicated in deregulating the cell cycle; however, the literature is conflicting and may reflect the non-physiological nature of many studies that used HBx overexpression systems.

In summary, our findings show that SAMHD1 can regulate multiple stages of the HBV replication cycle. Its diverse roles in dNTP metabolism, DNA repair pathways and innate immune activation has implications for the regulation of many viruses that rely on host metabolic activities to replicate.

# Materials and Methods

### Reagents, inhibitors, and antibodies

All tissue culture media and supplements, including fetal calf serum, were obtained from Invitrogen. All tissue culture plasticware were purchased from Sarstedt. Entecavir (ETV) was purchased from Sigma. IFN $\alpha$ was obtained from PeproTech. Myrcludex B was a kind gift from Stephan Urban. Deoxynucleosides dA (D8668) and dT (T1895) were purchased from Sigma, and dG (HY-17563) and dT (HY-17564) from MedChem. The following antibodies were used: mouse anti-SAMHD1 (ab67820; Abcam); rabbit anti-pSAMHD1 (Cell Signaling Technology), goat anti RRM2 (Santa Cruz), mouse anti-$\beta$ Actin (Sigma), and anti-mouse Alexa 488 (Life Technologies). The SAMHD1 CRISPR construct px458-SAM-sgRNA-1_2 was generated by the WIMM genome engineering service and guide RNAs designed as previously reported (Herold et al, 2017).

### Cell lines and viruses

HepG2-NTCP (K7) cells were cultured in DMEM supplemented with 100U/ml penicillin, 0.1 mg/ml streptomycin, 10% FCS, GlutaMAX and nonessential amino acids. Purified HBV was produced from HepAD38 cells as previously reported (Ko et al, 2018). rHBV reporter virus expressing Gaussia luciferase (rHBV-GLuc) was generated by stably transfecting HepG2 cells with a 1.3 HBV genome with a TTR promoter followed by a Gaussia luciferase gene (pRR_rHBV-TTR-Gaussia) along with a helper construct for expressing HBV polymerase and surface proteins (pRR_TTR-Polymerase-LMS-IRES-Puro) as previously reported (Protzer et al, 1999; Untergasser et al, 2006). Lentiviral vectors expressing Wt SAMHD1 or K11A mutant were produced as described (Schaller, Pollpeter et al, 2014b). Briefly, 293T cells were transfected using Fugene HD (Promega UK Ltd) and extracellular media harvested at 48 and 72 h post-transfection, filtered through a 0.45-$\mu$m syringe filter and stored at –80°C. SIV$_{mac}$ containing Vpx virus-like particles and VSV-G–pseudotyped HIV-1 NL4.3 were produced as previously described (Goujon et al, 2008).

### Generation of *Samhd1* KO cells

HepG2-NTCP cells were transfected with 1 $\mu$g of the px458-SAM-sgRNA-1_2 construct using Fugene HD (Promega UK Ltd) according

to the manufacturer's protocol. 48 h post-transfection, the cells were cloned by limiting dilution, expanded, and individual clones screened for SAMHD1 expression by Western blot.

### HBV infection

HepG2-NTCP cells were seeded on collagen-coated plasticware and treated with 2.5% DMSO for 3 d before infection. The cells were infected with HBV at an MOI of 200 genome equivalents per cell in the presence of 4% polyethylene glycol 8,000 for 24 h. Viral inoculum was removed at 24 h postinfection, the cells were washed three times with PBS and maintained in DMEM 10% FCS supplemented with 2.5% DMSO.

### Synchronized HBV infection

HepG2-NTCP cells were treated with 2.5% DMSO for 3 d before infection and the cells chilled on ice for 15 min before infecting with the chilled viral inoculum. The cells were incubated at 4°C for 1 h and then transferred to 37°C for 6 h. Non-internalized virus was removed by treating with trypsin for 3 min followed by three washes with PBS. The cells were harvested and HBV DNA measured by quantitative PCR (qPCR) as described below. As a control, the cells were pretreated with heparin (50 IU/ml) for 1 h to block HBV uptake as previously reported (Schulze et al, 2007).

### HBV genome transfer using an adenovirus vector

Ad-HBV-GFP was prepared in 293T cells as described (Sprinzl et al, 2001) and HepG2-NTCP cells infected at 80% confluency at an MOI of 20  Focus Forming Units (FFU)/cell. Transduced cells were monitored for their secretion of HBV DNA and RT-dependency confirmed by treating cells with entecavir (1 $\mu$M) (Fig S7).

### PCR quantification of HBV DNA and RNA

Total cellular DNA and RNA were extracted from HBV-infected cells using the All-Prep-DNA and RNA kit (QIAGEN). Selective PCR of cccDNA was performed as described (Ko et al, 2018). Briefly, extracted DNA was treated with 5U of T5 exonuclease (NEB) for 37°C for 30 min followed by heat inactivation at 95°C. Treated DNA samples were amplified in a SYBR green qPCR reaction (PCR Biosystems) (see Table S1 for primer sequences). For HBV DNA quantification, a 10-fold dilution series of an external HBV plasmid was used. HBV pgRNA was quantified from DNase-treated RNA extracts using a one-step reverse transcriptase qPCR (RT-qPCR) kit (Takyon). HBV primer probes (FAM) were used to amplify pgRNA, with primer probes for $\beta$ 2 microglobulin (VIC) used as an internal control in a multiplexed RT-qPCR reaction. To quantify extracellular HBV from infected cells, supernatants were first DNase-digested with 1U of DNase at 37°C for 30 min (Thermo Fisher Scientific) to remove any non-virion–associated HBV DNA. DNase was subsequently inactivated by the addition of 1 mM EDTA. A 2× lysis buffer containing 0.1 M Tris–HCl (pH7.4), 50 mM KCl, 0.25% Triton X-100, and 40% glycerol was added to the digested supernatants. Lysates were used in a qPCR reaction to detect HBV rcDNA using specific primers HBV4F and HBV4R (Tropberger et al, 2015). Copy numbers were

calculated using a dilution series of an external HBV plasmid as before. All qPCR reactions were carried out on a Roche 96 Light-Cycler (Roche).

### Southern blot detection of HBV DNA

To detect HBV DNA forms, including cccDNA, DNA from HBV-infected cells was isolated using an HIRT extraction protocol and separated by agarose gel as previously reported (Yan et al, 2012; Ko et al, 2018). DNA was transferred to a nylon membrane and hybridized with a digoxigenin-labeled full-length HBV probe. Luminescent signal was detected using the DIG Luminescent Detection Kit (Roche).

### SDS–PAGE and Western blots

The cells were lysed in RIPA buffer (20 mM Tris, pH 7.5, 2 mM EDTA, 150 mM NaCl, 1% NP40, and 1% sodium deoxycholate) supplemented with protease inhibitor cocktail tablets (Roche). 4× reducing buffer was added to samples before incubating at 95°C for 5 min. Proteins were separated on a 10% polyacrylamide gel and transferred to PVDF membranes (Amersham). The membranes were blocked in PBST, 5% skimmed milk (Sigma), and proteins detected using specific primary and HRP-secondary antibodies. Protein bands were detected using Pierce SuperSignal West Pico chemiluminescent substrate kit (Pierce) and images collected with a PXi Touch gel and membrane Imaging system (Syngene).

### Measurement of deoxynucleoside triphosphate levels in cells

Assays were performed as previously reported (Wilson et al, 2011). Briefly, reactions contained 100 pmol of NDP-1 primer (5′-CCGCCTCCACCGCC-3′), probe (5′-6FAM/AGGACCGAG/ZEN/GCAAGA-GCGAGCGA/IBFQ-3′), and 150 pmol of the template (5′-TCGCTCGC-TCTTGCCTCGGTCCT/BHQ-1/AGCGGCGGTGGAGGCG G-3′). Non-limiting dNTPs were added to a final concentration of 100 $\mu$M (except the dNTP to be measured). GoTaq Hot Start was added at a concentration of 0.5 U per reaction with 5× Colorless GoTaq Flexi Buffer and $MgCl_2$ at a final concentration of 1.5 mM. Reactions were made up to 10 $\mu$l with nuclease-free water, the volume of which was adjusted to incorporate 2 $\mu$l of cell extract. Reactions were performed on a Roche 96 LightCycler (Roche). The thermal profile consisted of 3 min at 95°C and a primer extension time of 64°C for 20 min. Raw fluorescence readings for the 6-FAM filter were measured every 30 s and analyzed using Microsoft Excel (Microsoft) and Prism 7 (GraphPad Software).

### Immunofluorescence

The cells were plated on glass coverslips, washed with PBS, and fixed with 4% paraformaldehyde for 20 min at room temperature. The coverslips were treated with 10 mM $NH_4Cl$ for 10 min at room temperature followed by PBS 0.1% (vol/vol) Triton-X for 5 min on ice. The coverslips were blocked, and incubated with the primary antibody (1:200) for 1 h at room temp, before washing and incubated with 488-conjugated secondary antibody (1:1,000). The coverslips were counterstained with 4′,6-diamidino-2-phenylindole and mounted with Prolong Gold Antifade (Invitrogen). The slides were viewed on a Zeiss fluorescence microscope (Carl Zeiss AG, DE).

### RPA32 staining

HepG2-NTCP cells were cultured on glass coverslips, treated with 1 $\mu$M CPT (Sigma) for 4 h, permeabilized with nuclear extraction buffer (10 mM PIPES, 20 mM NaCl, 3 mM MgCl$_2$, 300 mM sucrose, and 0.5% Triton X-100) for 5 min on ice before fixing in 3.6% para-formaldehyde for 10 min at room temperature. The cells were washed three times with PBS and blocked with ADB for 1 h at 4°C and incubated with anti-Replication Protein A2 (PA32) RPA32 (Cat. No. NA18; Millipore) and anti-P-RPA (S4/8) (Cat. No. A300-245A; Bethyl Laboratories) antibodies (diluted in ADB) for 1 h at room temperature. Unbound antibody was removed by washing with PBS and cells counterstained with Alexa Fluor–conjugated secondary antibodies (diluted in ADB). Finally, the cells were washed twice with ADB and coverslips mounted onto glass slides with Vectashield mounting agent containing 0.4 $\mu$g/ml DAPI (Vectashield). Images were taken using a Nikon Eclipse Ni microscope equipped with a 60× oil lens and at least 200 cells quantified per condition.

## Supplementary Information

## Acknowledgements

We acknowledge the technical assistance of Claudia Orbegozo Rubio and thank Philip Hublitz for help with generating *Samhd1* KO cells, Stephan Urban (University of Heidelberg) for providing Myrcludex B, and Barbara Testoni (INSERM, Lyon) for helpful discussions. JA McKeating laboratory is funded by EU 2020 Research and Innovation Programme Consortia HEP-CAR under grant agreement no. 667273, Wellcome Trust IA 200838/Z/16/Z, and MRC project grant MR/R022011/1. C Kramer was funded by a Lister Institute Summer studentship. M Higgs was funded by an MRC Career Development Fellowship (MR/P009085/1) and Birmingham Fellowship awarded by the University of Birmingham. J Rehwinkel laboratory was funded by the UK Medical Research Council (MRC core funding of the MRC Human Immunology Unit) and Wellcome Trust IA 100954. T Davenne was supported by the Wellcome Trust Infection and Immunology & Translational Medicine doctoral programme 105400/Z/14/Z. The WIMM Genome Engineering Facility is supported by grants from the MRC/MHU (MC-UU-12009), John Fell Fund (123/737), and by the WIMM Strategic Alliance awards G0902418 and MC_UU_12025. U Protzer laboratory is funded by the German Research Foundation via the collaborative research center TRR179 (project TP14; U Protzer). U Protzer and JA McKeating collaboration was supported by funding by the Institute for Advanced Study with the support of the Technical University of Munich via the German Excellence Initiative and EU 7th Framework Program under grant agreement no. 291763.

### Author Contributions

PAC Wing: conceptualization, data curation, formal analysis, writing—original draft, review, and editing.
T Davenne: resources.
J Wettengel: resources.
AG Lai: writing—review and editing.
X Zhuang: resources.
A Chakraborty: resources and data curation.
V D'arienzo: data curation.
C Kramer: resources, data curation, and supervision.
C Ko: resources, data curation, and supervision.
JM Harris: data curation and supervision.
S Schreiner: data curation, supervision, and writing—review and editing.
M Higgs: conceptualization, data curation, and writing—review and editing.
S Roessler: resources.
JL Parish: conceptualization, data curation, and writing—original draft, review, and editing.
U Protzer: conceptualization, resources, and writing—review and editing.
P Balfe: conceptualization, resources, data curation, software, formal analysis, supervision, funding acquisition, investigation, writing—original draft, project administration, and writing—review and editing.
J Rehwinkel: conceptualization, resources, and writing—review and editing.
JA McKeating: conceptualization, resources, data curation, software, formal analysis, supervision, funding acquisition, investigation, project administration, and writing—original draft, review, and editing.

### Conflict of Interest Statement

The authors declare that they have no conflict of interest.

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
