## [Reviewer comments · Life Science Alliance]

Life Science Alliance

A dual role for SAMHD1 in HBV cccDNA synthesis and RT-dependent particle genesis.

Peter Wing, Tamara Davenne, Jochen Wettengel, Alvina Lai, Xiaodong Zhuang, Anindita Chakraborty, Valentina D'Arienzo, Catharina Kramer, Chunkyu Ko, James Harris, Sabrina Schreiner, Martin Higgs, Stephanie Roessler, Joanna Parish, Ulrike Protzer, Peter Balfe, Jan Rehwinkel, and Jane McKeating

Corresponding author(s): Jane McKeating, University of Oxford

Review Timeline:	Submission Date:	2019-02-21
	Editorial Decision:	2019-02-21
	Revision Received:	2019-03-13
	Editorial Decision:	2019-03-15
	Revision Received:	2019-03-15
	Accepted:	2019-03-18

Scientific Editor: Andrea Leibfried

Transaction Report:

Please note that the manuscript was previously reviewed at another journal and the reports were taken into account in inviting a revision for publication at Life Science Alliance prior to submission to Life Science Alliance.

DOI: <https://doi.org/10.26508/lsa.201900355>

Referee #1 Review

Report for Author:

The manuscript by Wing et al addresses the role of the dNTPase SAMHD1, that is known to restrict HIV infection in quiescent cells and acts at stalled replication forks to prevent interferon induction, in infection of hepatocytes with hepatitis B virus. A previous study (Sommer et al 2016) reported that SAMHD1 had a restrictive influence on release of HBV from hepatocytes. Yet, this study only used a cell line with an HBV integrate leaving out the process of infection and employed RNAi to reduce SAMHD1 levels. The study under review here addresses the relevance of SAMHD1 for HBV infection using a cell line that can be infected with HBV and the authors employed both, a genetic SAMHD1 knockout using Crispr-Cas9 and a Vpx-based knock-down of SAMHD1 for their studies. Their results provide sound evidence for a previously unappreciated role of SAMHD1 in the nucleus for cccDNA formation using re-complementation with mutated SAMHD1 lacking an NLS. Using infection of HepG2-NTCP or delivery of HBV genomes by AdHBV they provide clear evidence for the restrictive function of SAMHD1 on release of HBV DNA-containing particles. Altogether, Wing et al identify SAMHD1 as host dependency as well as restriction factor for HBV that is explained by its distinct effects on the HBV life cycle.

The experiments are very well designed and the SAMHD1 knockout and re-complementation studies provide compelling evidence for its role as dependency and restriction factor for HBV. Some issues require attention

1. HBV cccDNA: the authors use Vpx-mediated depletion of SAMHD1 as well as SAMHD1 knockout cells to demonstrate that cccDNA formation is reduced in the absence of SAMHD1. A PCR-based detection method is employed in Figures 1 - 3 to quantify HBV cccDNA formation. Another method for HBV cccDNA detection beyond the PCR and luciferase reporter assay such as Southern Blot to confirm the results would substantially strengthen their conclusion. The experiment in Figure 1 also demonstrates that Vpx-mediated knock-down of SAMHD1 at later time points after infection does not influence HBV cccDNA levels any more. The authors conclude that this demonstrates that SAMHD1 is not involved in the maintenance of HBV cccDNA. Given the long half-life of cccDNA the authors need to rephrase their statement or provide more experimental evidence on any long-term effect of SAMHD1 on HBV cccDNA.
2. Using SAMHD1 knockout cells the authors demonstrate that HBV infection leads to induction of ISGs. This is also a very important finding, as HBV is considered a stealth virus that does not trigger innate immunity. More controls are needed to evaluate the importance of the increase in ISGs following HBV infection of SAMHD1 ko hepatocytes.

Minor point

- In Figure 5b, the authors show ETV as control but provide little information in the text as to why this is important.

Referee #2 Review

Report for Author:

In this manuscript, the authors describe two opposite roles of SAMHD1 during HBV infection: SAMHD1 positively regulates cccDNA levels and simultaneously negatively restricts the secretion of HBV DNA from infected cells. The study is well written and clear. The data is convincing to support the observations that in SAMHD1 KO cells, the levels of early cccDNA in the cell are decreased, and that paradoxically the levels of released viral DNA is increased. It is also convincing that the effect on early cccDNA levels requires SAMHD1 to be present in the nucleus but is independent of its dNTPase activity. These are interesting observations, but the underlying virological/immune/cellular mechanisms have not been defined. Furthermore, the restrictive role of SAMHD1 is not novel.

1. The restrictive role of SAMHD1 on HBV replication is already well established in several papers that are cited (Sommer 2016 Scientific Reports, Jeong 2016 Virology, Chen 2014 BBRC). The dNTPase-dependent mechanism of restriction of HBV and its regulation by CDK is well established (see also doi:10.1002/1873-3468.13105). The novelty here is the use of SAMHD1 KO cells or Vpx (figure 6) instead of siRNA, but this brings no conceptual or mechanistic advance. They argue that the fold-increase with KO cells is unprecedented, but there is no mechanistic insight to validate that this is the result of gene editing as opposed to partial reduction with Vpx, and the relevance remains limited to immortalized HepG2 cells. Given the critical role of SAMHD1 in DDR in immortal cancer cells, they should knock-out SAMHD1 in primary hepatocytes instead to bring novelty.

2. Second, since it is clear that the overall effect on SAMHD1 on HBV production by cells is a restrictive role, the significance of the positive role on SAMHD1 on cccDNA levels is unclear, and the overall interpretation that SAMHD1 would positively repair the cccDNA is undemonstrated. They observe a decrease in early levels of cccDNA in SAMHD1 KO cells (and consequently a similar decrease in viral RNA levels, Fig S4) - but the level of secreted DNA is instead increased. It is very difficult to understand how decreased cccDNA levels in KO cells would lead to an increase in viral replication and viral DNA release. It makes more sense that the difference in DNA levels is an epiphenomenon of SAMHD1 on cccDNA, and that the important effect SAMHD1 is instead to qualitatively modify the cccDNA in WT cells vs KO cells. This would fit better with the Coquel et al. study that demonstrates a role of SAMHD1 in DNA repair. A number of virological mechanisms can account for how an altered cccDNA state by SAMHD1, at the cost of a slight increase in cccDNA levels, could interfere with overall viral replication, and unfortunately, they have not performed the required detail virological characterization of viral DNA/RNA species, viral protein expression, assembly, budding, release and infectivity of the released virus to understand this.

3. Third, the mechanism by which SAMHD1 would promote early cccDNA levels is undefined. Their final message on this is that it is dNTPase-independent, requires nuclear localization and is associated with an increase in interferon-stimulate gene expression upon infection. These are interesting observations, but the mechanism remains undetermined. They have not examined the roles of Mre11, ATR, CHK1, cGAS, STING, IFN, IFNAR, ISGs, etc. There is also no direct demonstrate that SAMHD1 would promote cccDNA "synthesis" or "genesis" (as claimed throughout the text and title) - it is equally possible that SAMHD1 prevents its degradation.

Other comments:

- The use of statistical analysis to support n=2 experiments, that are numerous in this study, is questionable
- The conclusion in the discussion that SAMHD1 is required to repair viral DNA and to inhibit reverse transcription is largely an over-interpretation of very indirect assays that only measure levels of cccDNA in cells and secreted viral DNA.

February 21, 2019

Re: Life Science Alliance manuscript #LSA-2019-00355-T

Prof. Jane A. McKeating
University of Oxford

Dear Dr. McKeating,

Thank you for transferring your manuscript entitled "A dual role for SAMHD1 in HBV cccDNA synthesis and RT-dependent particle genesis" to Life Science Alliance. The manuscript was assessed by expert reviewers at another journal before, and the editors transferred those reports to us with your permission.

The reviewers thought that your work is robust but they expected further reaching mechanistic insight. The lack thereof is not a concern for publication in Life Science Alliance, and we would thus like to invite you to provide a slightly revised version of this work for publication here. Reviewer #2 comments (points 2 and 3) should get addressed in the manuscript text by discussing the current knowledge gaps and by mentioning alternative hypotheses. It would be also good to add an alternative assay for measuring cccDNA (eg Southern Blot as suggested by rev#1). The minor point of this reviewer should get addressed and please discuss your results of figure 4 more broadly to address point #2 of this reviewer.

The typical timeframe for revisions is three months. Please note that papers are generally considered through only one revision cycle.

Thank you for this interesting contribution to Life Science Alliance. We are looking forward to receiving your revised manuscript.

Sincerely,

Andrea Leibfried, PhD
Executive Editor

Life Science Alliance
Meyerhofstr. 1
69117 Heidelberg, Germany
t +49 6221 8891 502
e a.leibfried@life-science-alliance.org
www.life-science-alliance.org

B. MANUSCRIPT ORGANIZATION AND FORMATTING:

NDM Research Building

12th March 2019

Dr Andrea Leibfried, Executive Editor
Life Science Alliance

Re: Life Science Alliance manuscript #LSA-2019-00355-T

Dear Andrea

Please find attached our responses to your editorial feedback requesting a revised version of our manuscript along with a detailed point-by-point response to referee comments from our earlier EMBO J submission.

Life Science Alliance request:

Reviewer #2 comments (points 2 and 3) should get addressed in the manuscript text by discussing the current knowledge gaps and by mentioning alternative hypotheses. **Response:** We have revised text accordingly.

It would be good to add an alternative assay for measuring cccDNA (eg Southern Blot as suggested by rev#1). **Response:** We have provided Southern blots in Fig.1 and Fig.2.

The minor point of this reviewer should get addressed and please discuss your results of figure 4 more broadly to address point #2 of this reviewer. **Response:** We have discussed the use of ETV and provide a new supplementary Fig.5 in response to this minor query.

Referee #1: The manuscript by Wing et al addresses the role of the dNTPase SAMHD1, that is known to restrict HIV infection in quiescent cells and acts at stalled replication forks to prevent interferon induction, in infection of hepatocytes with hepatitis B virus. A previous study (Sommer et al 2016) reported that SAMHD1 had a restrictive influence on release of HBV from hepatocytes. Yet, this study only used a cell line with an HBV integrate leaving out the process of infection and employed RNAi to reduce SAMHD1 levels. The study under review here addresses the relevance of SAMHD1 for HBV infection using a cell line that can be infected with HBV and the authors employed both, a genetic SAMHD1 knockout using Crispr-Cas9 and a Vpx-based knock-down of SAMHD1 for their studies. Their results provide sound evidence for a previously unappreciated role of SAMHD1 in the nucleus for cccDNA formation using re-complementation with mutated SAMHD1 lacking an NLS. Using infection of HepG2-NTCP or delivery of HBV genomes by AdHBV they provide clear evidence for the restrictive function of SAMHD1 on release of HBV DNA-containing particles. Altogether, Wing et al identify SAMHD1 as host dependency as well as restriction factor for HBV that is explained by its distinct effects on the HBV life cycle.

The experiments are very well designed and the SAMHD1 knockout and re-complementation studies provide compelling evidence for its role as dependency and restriction factor for HBV. Some issues require attention

1. HBV cccDNA: the authors use Vpx-mediated depletion of SAMHD1 as well as SAMHD1 knockout cells to demonstrate that cccDNA formation is reduced in the absence of SAMHD1. A PCR-based detection method is employed in Figures 1 - 3 to quantify HBV cccDNA formation. Another method for HBV cccDNA detection beyond the PCR and luciferase reporter assay such as Southern Blot to confirm the results would substantially strengthen their conclusion. **Response:** We provide Southern blots to confirm effect of Vpx-KD (Fig.1) and SAMHD1 KO (Fig.2) on HBV cccDNA levels in the revised manuscript.

The experiment in Figure 1 also demonstrates that Vpx-mediated knock-down of SAMHD1 at later time points after infection does not influence HBV cccDNA levels any more. The authors conclude that this demonstrates that SAMHD1 is not involved in the maintenance of HBV cccDNA. Given the long half-life of cccDNA the authors need to

rephrase their statement or provide more experimental evidence on any long-term effect of SAMHD1 on HBV cccDNA. **Response:** We have amended the text to provide an alternative explanation for the results from Fig.1 and discuss the long half-life of cccDNA.

2. Using SAMHD1 knockout cells the authors demonstrate that HBV infection leads to induction of ISGs. This is also a very important finding, as HBV is considered a stealth virus that does not trigger innate immunity. More controls are needed to evaluate the importance of the increase in ISGs following HBV infection of SAMHD1 KO hepatocytes. **Response:** We are currently assessing the STING dependency of this immune activation along with sequencing extracellular HBV DNA produced from Wt and SAMHD1 KO cells to probe for APOBEC mediated deamination signatures. We feel these experiments are beyond the remit of the current MS.

Minor point: In Figure 5b, the authors show ETV as control but provide little information in the text as to why this is important. **Response:** We have amended the text to provide a detailed description of ETV and included new data showing the kinetics of ETV treated HepG2.2.15 cells on HBV DNA extracellular levels and how drug removal enables one to follow de novo particle secretion.

Referee #2: In this manuscript, the authors describe two opposite roles of SAMHD1 during HBV infection: SAMHD1 positively regulates cccDNA levels and simultaneously negatively restricts the secretion of HBV DNA from infected cells. The study is well written and clear. The data is convincing to support the observations that in SAMHD1 KO cells, the levels of early cccDNA in the cell are decreased, and that paradoxically the levels of released viral DNA is increased. It is also convincing that the effect on early cccDNA levels requires SAMHD1 to be present in the nucleus but is independent of its dNTPase activity. These are interesting observations, but the underlying virological/immune/ cellular mechanisms have not been defined. Furthermore, the restrictive role of SAMHD1 is not novel.

1. The restrictive role of SAMHD1 on HBV replication is already well established in several papers that are cited (Sommer 2016 Scientific Reports, Jeong 2016 Virology, Chen 2014 BBRC). The dNTPase-dependent mechanism of restriction of HBV and its regulation by CDK is well established (see also doi:10.1002/1873-3468.13105). The novelty here is the use of SAMHD1 KO cells or Vpx (figure 6) instead of siRNA, but this brings no conceptual or mechanistic advance. They argue that the fold-increase with KO cells is unprecedented, but there is no mechanistic insight to validate that this is the result of gene editing as opposed to partial reduction with Vpx, and the relevance remains limited to immortalized HepG2 cells. Given the critical role of SAMHD1 in DDR in immortal cancer cells, they should knock-out SAMHD1 in primary hepatocytes instead to bring novelty. **Response:** We believe our data showing a 10-100 fold increase in the rate of HBV particle secretion from SAMHD1 KD or KO cells shows a more robust phenotype than previously published work. To provide mechanistic insight between our current study and previous reports in the literature, we provide new data in the revised MS showing that the SAMHD1 KD phenotype in persistent HBV producer lines is only apparent when the cells are pre-treated with the nucleoside analog Entecavir before LVP-Vpx transduction. These data support a model where pre-formed intracellular HBV encapsidated rcDNA is insensitive to changes in dNTP levels and their long half-life may limit the apparent effectiveness of agents targeting RT-dependent step in the viral life cycle.

2. Second, since it is clear that the overall effect on SAMHD1 on HBV production by cells is a restrictive role, the significance of the positive role on SAMHD1 on cccDNA levels is unclear, and the overall interpretation that SAMHD1 would positively repair the cccDNA is undemonstrated. They observe a decrease in early levels of cccDNA in SAMHD1 KO cells (and consequently a similar decrease in viral RNA levels, Fig S4) - but the level of secreted DNA is instead increased. It is very difficult to understand how decreased cccDNA levels in KO cells would lead to an increase in viral replication and viral DNA release. **Response:** We agree that the overall effect of SAMHD1 is to restrict *de novo* particle genesis. However, we respectively disagree that a positive role for SAMHD1 in regulating early steps of HBV cccDNA genesis is incompatible with a restrictive role later in the viral life cycle. Indeed, our data suggests a role for SAMHD1 phosphorylation to regulate the different steps in the life cycle and suggest HBV de-regulation of cell cycle as a potential mechanism in the discussion. It makes more sense that the difference in DNA levels is an epiphenomenon of SAMHD1 on cccDNA, and that the important effect SAMHD1 is instead to qualitatively modify the cccDNA in WT cells vs KO cells. This would fit better with the Coquel et al. study that demonstrates a role of SAMHD1 in DNA repair. A number of virological mechanisms can account for how an altered cccDNA state by SAMHD1, at the cost of a slight increase in cccDNA levels, could interfere with overall viral replication, and unfortunately, they have not performed the required detail virological characterization of viral DNA/RNA species,

viral protein expression, assembly, budding, release and infectivity of the released virus to understand this. **Response:** If SAMHD1 regulates cccDNA epigenome one would expect to see differences in the transcriptional activity of the DNA, and this is not supported by our data showing comparable levels of pre-genomic RNA in Wt and KO cells relative to cccDNA levels (Supplementary Fig.4). We have revised text to discuss the HBV epigenome.

3. Third, the mechanism by which SAMHD1 would promote early cccDNA levels is undefined. Their final message on this is that it is dNTPase-independent, requires nuclear localization and is associated with an increase in interferon-stimulated gene expression upon infection. These are interesting observations, but the mechanism remains undetermined. They have not examined the roles of Mre11, ATR, CHK1, cGAS, STING, IFN, IFNAR, ISGs, etc. There is no direct demonstration that SAMHD1 would promote cccDNA "synthesis" or "genesis" (as claimed throughout the text and title) - it is equally possible that SAMHD1 prevents its degradation. **Response:** At the present time the pathways regulating cccDNA genesis are not well defined in the literature and we believe our data provides convincing evidence for a role for SAMHD1 in early steps of cccDNA formation. cccDNA is long lived and so the timing of our experiments within 3 days post infection we would be unlikely to detect cccDNA degradation. We have edited the text to discuss this possible explanation of our data.

We look forward to a favourable review,

With best wishes,

Jane A McKeating

Professor of Molecular Virology

March 15, 2019

RE: Life Science Alliance Manuscript #LSA-2019-00355-TR

Prof. Jane A. McKeating
University of Oxford
Old Road Campus
Roosevelt Drive,
Oxford OX3 7FZ
United Kingdom

Dear Dr. McKeating,

Thank you for submitting your revised manuscript entitled "A dual role for SAMHD1 in HBV cccDNA synthesis and RT-dependent particle genesis". I appreciate the introduced changes and would be happy to publish your paper in Life Science Alliance.

Before sending you an official acceptance letter, please log in one more time to fill in the electronic license to publish form. Your manuscript number will change to LSA-2019-00355-TRR, please make sure to move all manuscript files to this new number (single click process). Please also link your profile to your ORCID iD (you should have received an email with instructions on how to do so) and check whether all author contributions meet ICMJE authorship guidelines.

A. FINAL FILES:

-- Summary blurb (enter in submission system): A short text summarizing in a single sentence the study (max. 200 characters including spaces). This text is used in conjunction with the titles of papers, hence should be informative and complementary to the title. It should describe the context

and significance of the findings for a general readership; it should be written in the present tense and refer to the work in the third person. Author names should not be mentioned.

B. MANUSCRIPT ORGANIZATION AND FORMATTING:

Sincerely,

March 18, 2019

RE: Life Science Alliance Manuscript #LSA-2019-00355-TRR

Prof. Jane A. McKeating
University of Oxford
Old Road Campus
Roosevelt Drive,
Oxford OX3 7FZ
United Kingdom

Dear Dr. McKeating,

Thank you for submitting your Research Article entitled "A dual role for SAMHD1 in HBV cccDNA synthesis and RT-dependent particle genesis.". It is a pleasure to let you know that your manuscript is now accepted for publication in Life Science Alliance. Congratulations on this interesting work.

DISTRIBUTION OF MATERIALS:

Again, congratulations on a very nice paper. I hope you found the review process to be constructive and are pleased with how the manuscript was handled editorially. We look forward to future exciting submissions from your lab.